

# Hydrometeorological analysis of the 12 and 13 September 2019 widespread flash flooding in eastern Spain

Arnau Amengual

Grup de Meteorologia, Departament de Física, Universitat de les Illes Balears, Palma, Mallorca, Spain.

*Correspondence to*: Arnau Amengual (arnau.amengual@uib.es)

**Abstract.** On 12 and 13 September 2019, a long-lasting heavy precipitation episode (HPE) affected the València, Murcia and Almería regions in eastern Spain. Observed rainfall amounts were close to 500 mm in 48 h, being the highest cumulative precipitation registered in some rain-gauges for the last century. Subsequent widespread flash flooding caused seven fatalities

and estimated economical losses above 425 million EUR. High-resolution precipitation estimates from weather radar observations and flood response from stream-gauges are used in combination with a fully-distributed hydrological model to examine the main hydrometeorological processes within the HyMeX program. This HPE was characterized by successive, well-organized convective structures that impacted a spatial extent of 7500 km$^2$, with rainfall amounts equal or larger than 200 mm. The main factors driving the flood response were quasi-stationarity of heavy precipitation, very dry initial soil moisture

conditions and large storage capacities. Most of the examined catchments exhibited a dampened and delayed hydrological response to cumulative precipitation: Until runoff thresholds were exceeded, infiltration-excess runoff generation did not start. This threshold-based hydrological behaviour may impact the shape of flood peak distributions, hindering strict flood frequency statistical analysis due to the generally limited lengths of data records in arid and semi-arid catchments. As an alternative, simple scaling theory between flood magnitude and total rainfall amount is explored.

## 1 Introduction

Flash floods are among the most devastating natural hazards in terms of economic losses and death toll worldwide (e.g., CRED 2016; Petrucci et al. 2019). In addition, the potential of flash flood-related casualties and damages is steadily increasing in many regions due to the synergies between social and economic pressures on land use and global warming. Among other scientific goals, the Hydrological Cycle in the Mediterranean Experiment (HyMeX; http://www.hymex.org) program aims at

a better understanding, modelling and forecasting of hydrometeorological extremes over the flood-prone Mediterranean region in an era of climate change (Drobinski et al. 2014).

Mediterranean Spain is impacted by heavy precipitation episodes (HPEs) that result in hazardous flash floods every year. During late summer and early autumn, the relatively high sea surface temperature is a source of heat and moisture for low-level air masses. Strong convective instability is fostered by the early entrance of mid-level cold disturbances. Subsequent

deep convective systems are often anchored by the prominent orography of the region, resulting in quasi-stationary heavy



rainfall over specific watersheds (Romero et al. 2000; Llasat et al. 2003; García-Herrera et al. 2005; Martín et al. 2007; Pastor et al. 2010; Hermoso et al., 2021).

The complex and steep orography of the Spanish Mediterranean region is nearby the coast, shaping numerous small-to-medium, semi-arid basins (Fig. 1). These ephemeral catchments feature hydrological responses to extreme precipitation of only a few hours, reacting with an acute spatial and temporal variability. On the one hand, deep convection can lead to large rainfall rates and accumulations with an intrinsically high heterogeneity in space and time. On the other hand, thin soils, sparse vegetation, abrupt slopes and urban development can result in fast Hortonian runoff generation, overland and channel flows. After the long dry and warm summer, extreme rainfall rates can certainly overwhelm the initially large soil infiltrabilities, resulting in sudden flood bores routing through normally dry river beds with catastrophic effects downstream (Amengual et al. 2007 and 2015; Martín-Vide and Llasat 2018; Lorenzo-Lacruz et al. 2019).

Rainfall amount can also be an important flash flood-triggering factor in arid and semi-arid watersheds: Until runoff thresholds are exceeded, sudden infiltration-excess runoff generation is not triggered (Smith et al. 1996; Camarasa-Belmonte and Beltrán Segura 2001; Gaume et al. 2003; Delrieu et al. 2005). This threshold-type hydrological behaviour leads to a belatedly but then sudden runoff production. The catchment reacts as a system that becomes progressively saturated or as a tip-over system, with a sudden rise of runoff coefficients. Nonlinearities related to this wetting-up process result in even greater spatial heterogeneity than the conducive rainfall (Gaume et al. 2004; Borga et al. 2007). Admittedly, the physiographic and morphological properties of the Spanish Mediterranean region are also dominant factors when modulating complexity in hydrological response. Besides very dry initial soil moisture conditions, many catchments lie on very fractured and karstified bedrock. These substrates favour large water storage capacities that recharge deep aquifers via infiltration, percolation and transmission losses (Camarasa-Belmonte and Beltrán Segura 2001; Camarasa-Belmonte 2016; Amengual et al. 2017).

A better understanding of the impact of the geomorphological characteristics of arid and semi-arid catchments under flash flood conditions could help to build up more homogeneous datasets for regional frequency studies. In Spain, hydrological design and flood risk management rely on estimating quantiles of peak discharges characterized by low annual exceedance probabilities. That is, flood frequency analysis is carried out by identifying which analytical statistical distributions reproduce better the cumulative distribution functions of extreme flood peaks based on long-term observations of annual maximum discharges (Álvarez et al., 2012). The better fit is mainly observed for the generalized extreme value distribution, which is also extensively used for flood frequency analysis in many other regions worldwide (Metzger et al., 2020). However, dense observational networks with long-term discharge series are scarce in arid and semi-arid basins. When flow records are not long enough to perform a strict statistical analysis or basins are ungauged, multiple regression analysis or hydrometeorological simulations can be used to transfer flood quantiles from catchments with long-term data records to watersheds with hydrological similarity. This procedure aims at providing a continuous and useful mapping of flood risks for civil protection decision-makers.

Admittedly, hydrological response based on a threshold-type behaviour may affect the shape of flood peak distributions. Arid and semi-arid watersheds feature a very limited number of floods per year, but exhibit a large variability in peak discharge as



different rainfall rates and amounts trigger floods with uneven specific peak discharges (Gaume et al. 2004). Therefore, it is difficult to extrapolate flood quantile data from well-monitored basins to similar catchments with limited observations. Some authors have pointed out that although rainfall-runoff processes governing flash floods in arid and semi-arid basins such as, convectively-driven precipitation, infiltration-excess runoff and surface water movement in hillslopes and channels, are highly variable in space and time, the aggregated behaviour of peak flows exhibits certain scale invariances with the parameters

representing catchment and rainfall properties (e. g., Gupta and Dawdy 1995; Ogden and Dawdy 2003; Gupta 2004; Furey and Gupta 2005 and 2007; Gaume et al. 2009; Marchi et al. 2010; Schumer et al. 2014).

Some of these works have delved into scale-invariant behaviours that allow to predict peak magnitude as an alternative to flood frequency analysis when long-term stream flow records are not available. Ogden and Dawdy (2003) observed that flood peak quantiles exhibit self-similarity with catchment area on an event-by-event basis for a small watershed with Hortonian

runoff response to convective precipitation. Schumer et al. (2014) showed the existence of an empirical power-law relating peak discharge to direct runoff volume for a given event duration, after examining long-term flow series in 28 arid and semi-arid basins across the south-western United States, with areas ranging from 2 to 1256 $km^2$. Both studies empirically-derived envelope curves so as to appraise the upper bound of flood magnitudes depending on the selected physical variable. Envelope curves have the advantage of being relatively unaffected by data limitations as they are determined by the maximum values of

a sample.

The heaviest precipitation took place on 12 and 13 September: 10-min rainfall amounts were above 25 mm in several automatic rain-gauges, with a maximum 48-h observed amount of 492 mm. Subsequent widespread flash flooding devastated wide areas of the València, Murcia and Almería regions in eastern Spain (Fig. 1): the death toll was 7, hundreds of dwellings were evacuated and economic losses were estimated above EUR 425 million (CCS 2019). The 12-13 September 2019 HPE

represents a prototype in terms of well-organized and sustained convective systems leading to the most catastrophic flash floods over the Spanish Mediterranean area. In line with previous flash flood monographs (e.g., Smith el at. 1996 and 2000; Ogden et al. 2000; Gaume et al. 2003 and 2004; Delrieu et al. 2005; Borga et al. 2007, Zanon et al. 2010), the first objective is to examine the main rainfall and runoff processes that concurred in the unfolding of this extreme episode. At this aim, high-resolution quantitative precipitation estimates (QPEs) derived from weather radar observations and flood measurements from

automatic stream-gauges are used together with a fully-distributed hydrological model.

A remarkable characteristic of the 12-13 September 2019 extreme episode was the dampened and delayed hydrological response to heavy precipitation of most catchments examined, pointing out cumulative precipitation as another important flash flood-triggering mechanism. Therefore, the second objective is to further explore simple scaling theory between flood magnitude and rainfall amount for this study case. In addition, the large spatial extension of this long-lasting HPE together

with the contrasting physiography and morphology of the selected basins also permits cross-validations and inter-comparisons among their hydrological responses (Fig. 2). Both objectives are framed within the main scientific goals of the heavy rainfall, flash floods and floods section of HyMeX.



## 2 Study region and data

### 2.1 Selected catchments: an overview

The region of interest extends over 36700 km$^2$, dominated by the Baetic mountainous system (Fig. 1). This highly rugged relief is very close to the coast, directly sinking into the Mediterranean Sea. As a result, many small-to-medium ephemeral catchments –called "Ramblas" in eastern Spain– spread over the selected areat. Annual rainfall amounts are characterized by a decrease from north to south, passing from nearly 700 mm to less than 300 mm (Fig. 2). Conversely, annual mean temperature increases from 16 ℃ in the north to 18 ℃ in the south. According to the Köppen-Geiger classification, the dominant climate

is either hot or cold semi-arid (BSh/BSk), although the northernmost part is categorized as hot- and dry-summer Mediterranean (CSa) while some limited areas of the Almería coast are classified as hot desert (BWh; Chazarra-Bernabé et al. 2018).

The northernmost *Cànyoles* watershed settles over extensive karstified limestone, carbonate, and dolomitic fractured bedrock, favouring the recharge of deep calcareous aquifers (Sese-Minguez et al. 2017). The *Rambla Salada* basin lies on bedrock weakness primarily made up of marls in a badland area. As being located in a semi-arid climate, the long and recurrent drought

periods produce the generation of a dense network of desiccation cracks on the surface. As consequence, the hydrological response of the *Rambla Salada* is strongly modulated by: (i) exacerbating infiltration rates, and (ii) stemming overland flow (Cerdà 1995). The *Rambla del Albujón* features deep and silty soils with low perviousness (Fig. 2). However, current intensive agricultural practices have notably increased soil drainage and water storage (García-Pintado et al. 2009).

Soil profiles are profound and have medium to high permeability over the *Rambla de Benipila*. Nowadays, its eastern part is

also devoted to intensive agricultural production. This watershed lies over marble, mica schist, quartzite, gneiss, phyllite, and plaster bedrock. The *Rambla de Canalejas* is mainly located on dendritic and quaternary formations with medium-to-high perviousness. Other portions of the drainage area lie on mica schist, quartzite, phyllite, and plaster bedrock (IGME 1993). The mountainous headwaters of the *Almanzora* catchment are also settled over carbonate, limestone and dolomitic fractured bedrock, leading to large infiltrabilities that feed deep aquifers (Vallejos et al. 1994). As being hydraulically disconnected of

the underlying aquifers, all these river basins feature very irregular regimes. Most of their tributaries are dry and hydrologically active during flash floods. Note that for discussion purposes the *Ramblas Salada, del Albujón* and *de Benipila* are grouped as the central catchments. Accordingly, the *Rambla de Canalejas* and *Almanzora* watersheds are labelled as the southern basins (Fig. 2).

### 2.2 Quantitative precipitation estimates

QPEs are derived from the reflectivity volume scans of the Almería, Murcia and València Doppler C-band radars from 11 to 14 September at 00 UTC (Fig. 1). Spatial resolution is of 1-km in range and 0.8º in azimuth. A complete volume-scanning is performed every 10-min, with a maximum range of 240 km. Volume-scanning is affected by the complex orography of the region. Therefore, partial beam occlusion is amended by numerically simulating the beam power percentage blocked by the rugged topography. This amelioration is carried out by numerically modelling beam propagation over a high-resolution digital





terrain model (Pellarin et al. 2002). Signal attenuation by heavy rain is also corrected by means of the Mountain Reference

Technique (Bouilloud et al. 2009). Next, quantitative rainfall estimations are obtained by applying the standard WSR-88D

convective rainfall rate-reflectivity relationship (i.e., $Z = 300R^{1/4}$; Hunter 1996; Fulton et al. 1998).

As radar quantitative precipitation estimation entails large uncertainties (e.g., Gochis et al. 2015), additional inaccuracies in

the hourly cumulative rains and patterns are amended by using a dynamical fitting to the 369 automatic pluviometers available

over the region of interest (Fig. 1; Cole and Moore 2008). Finally, an independent safety check is performed by comparing the

48-h radar-derived precipitation against observations by 227 independent daily pluviometers. Statistical comparison between

both databases exhibits a strong positive correlation ($R^2 = 0.88$; Fig. 3a), even if QPEs feature a slight mean underestimation

of 11.1%.

**3 Precipitation analyses**

**3.1 Spatial and temporal distribution of rainfall at large scale**

According to the 48-h radar-derived rainfall, maximum cumulative precipitation is well above 200 mm in all the concerned

basins (Fig. 3b). The highest rainfall amounts are located in the *Cànyoles* and *Almanzora* catchments, where the lifting

associated with topographic forcing enhanced precipitation, with values well above 500 mm. In terms of the area sizes over

which a given amount of rainfall was exceeded, the spatial extents above the 200-, 400- and 600-mm thresholds were roughly

of 7500, 375 and 20 km², respectively. A striking characteristic of this HPE is the vast extension of the precipitation amounts

equal to or greater than 200 mm, as a consequence of the convection activity being exceptionally active and long-lasting.

Hermoso et al. (2021) divided the 12-13 September 2019 episode in the following three distinct phases:

- Phase 1 (00-06 UTC 12 Sep.): The spatial signature of the thin convective band responsible of heavy rainfall over the
northern part of the region of interest is clearly visible in the 6-h cumulative radar-derived rainfall amounts (Fig. 4a).
    Torrential precipitation exhibited maximum 10-minutes and 6-h accumulations of 16 mm and 210 mm, respectively.
    This persistent and elongated convective system produced extreme rainfall that lasted for approximately 6 h.

- Phase 2 (06-18 UTC 12 Sep.): A linear structure of convection resulted in intense rainfall over the central part of the
domain. This prominent convective system brought a maximum precipitation of 180 mm in 6 hours, whereas peak
    10-minute intensity was above 20 mm, according to the automatic pluviometric stations. In the afternoon, moderate
    rainfall rates persisted over southern València and western Murcia and Almería (Fig. 4b-c).

- Phase 3 (19 UTC 12 Sep.-12 UTC 13 Sep.): The most intense convective activity formed during this phase, including
a quasi-stationary and V-shaped mesoscale convective system (MCS). During its mature stage, the coastal lands of



Murcia were affected by hourly intensities up to 146 mm and 10-minute recordings exceeding 30 mm. In western Almería, rainfall-band structures led to maximum 6-h amounts above 200 mm (Figs. 4d-f).

Hermoso et al. (2021) also examined the main torrential features of this HPE over the studied basins. To this end, this work explored: (i) the link between maximum 30-min rainfall rates and cumulative precipitation and, (ii) the percentage of total rainfall amount with 30-min rates > 20 mmh⁻¹. The northern and central catchments were impacted by the strongest and lasting rains during phases 1 and 3. The large values of the total precipitation percentage exhibiting large intensities revealed the quasi-stationary aspect of the convective bands. Another striking property is that some automatic rain gauges were affected by major convective precipitation for most of the complete duration of the 12-13 September 20129 HPE. That is, eight pluviometers measured the 60 percent of rainfall amounts above 300 mm under heavy rainfall conditions over the northern and central watersheds. The southern basins were also impacted by extreme rainfall rates, but short-lived.

The different spatial and temporal scales of strong convective activity resulted in a self-similar organization between maximum 30-min rainfall rates and total precipitation amounts over the region (Fig. 5). This scale-invariant behaviour is related to the spatial structure, temporal evolution, orographic enhancement, and quasi-stationarity of the rainfall patterns. It is also likely that particular rainfall organizations, in which long-lasting convective clusters of very high rainfall rate were embedded in not so persistent convective clusters of lower precipitation intensity, which were, in turn, embedded in still shorter convective cells of even lower rainfall rate, contribute to self-similarity. After applying an ordinary least-squares (OLS) regression, the scaling intercept and exponent are of 1.02 $[mm^{1-b}h^{-1}]$ and 0.72, respectively, with a squared correlation coefficient of 0.86 (Fig. 5). The statistical significance of the OLS regression model is checked by means of the p-value, while uncertainty is quantified by using a non-parametric bootstrapping method (Hall et al., 2004). To this end, 1000 samples with replacement among the data are performed. Both tests confirm the robustness of the scaling relationship between maximum 30-min rainfall intensity and cumulative precipitation for the 12-13 September 2019 HPE, with a p-value less than 0.01. Furthermore, the 95% confidence interval of the reduced rainfall rate is of (0.95–1.01) $[mm^{1-b}h^{-1}]$, while for the scaling exponent is of (0.69–0.75).

## 3.2 Spatial and temporal distribution of rainfall at catchment scale

The role of the spatial and temporal variability of rainfall in flood response is measured by means of the spatial moments of catchment rainfall (Zoccatelli et al. 2011). The spatial moments detail the rainfall organization over the basin in terms of concentration and dispersion statistics, as a function of the distance measured along the flow path. Next, the mathematical formulation of the spatial moments of catchment rainfall is briefly revised.

The flow distance (d(x,y)) is defined as the path length from a given position, (x,y) to the catchment outlet. The n-th spatial moment of the rainfall field, $(r(x, y, t))$ is introduced as:

$$p_n(t) = |A|^{-1} \int_A r(x, y, t) \cdot d(x, y)^n \, dA, \tag{1}$$





where A stands for the drainage area and t for time. The zero-th order spatial moment ($p_0(t)$) renders the average catchment
rainfall intensity at time t. Likewise, the n-order moment of the flow distance ($g_n$) is introduced as:

$$g_n = |A|^{-1} \int_A d(x,y)^n \, dA. \tag{2}$$

The first two orders of the non-dimensional spatial moments of catchment rainfall are given by:

$$\delta_1(t) = \frac{1}{g_1}\left[\frac{p_1(t)}{p_0(t)}\right], \tag{3}$$

$$\delta_2(t) = \frac{1}{g_2 - g_1^2}\left[\frac{p_2(t)}{p_0(t)} - \left(\frac{p_1(t)}{p_0(t)}\right)^2\right] \tag{4}$$

The first-order scaled moment ($\delta_1(t)$) measures the distance between the catchment rainfall and basin centroids. Values of
$\delta_1(t)$ close to 1 indicate either a rainfall distribution located nearby the position of the watershed centroid or spatially uniform.
$\delta_1(t)$ smaller (greater) than unity reflect that rainfall is distributed near the basin outlet (headwaters). The second-order scaled
moment ($\delta_2(t)$) accounts for the dispersion of the precipitation-weighted flow distances about their mean value with respect
to the scattering of the flow distances. Values of $\delta_2(t)$ close to the unity point out to a uniform-like precipitation pattern, with
values less (greater) than 1 indicate that the rainfall pattern features a unimodal (multimodal) distribution along the flow
distance (Zoccatelli et al. 2011).

The temporal sequence of the spatial arrangement in precipitation with respect to catchment morphology is also assessed (Figs.
6 and 7). To this end, the hourly time series of the basin-area average precipitation, the fractional coverages of each catchment
impacted by hourly rainfall rates larger than 20 mmh$^{-1}$ and, the first and second spatial moments are computed for the selected
watersheds. Note that the fraction of a basin affected by intense downpours links the precipitation and watershed scales, being
an essential promoter to flood response (Smith et al. 2002; Veldhuis et al. 2018).

The northernmost *Cànyoles* basin was impacted by the thin and persistent convective band during phase 1 (Fig. 4a). In
consequence, the hourly catchment-area average rains were above 15 mm from 05 to 08 UTC on September 12, while the
fractional basin coverage of the heaviest hourly precipitation was slightly above 0.6, affecting an area somewhat larger than
525 km$^2$ (Figs. 6a-b). $\delta_1$ reflects that the torrential precipitation was mainly distributed close to the basin outlet (Fig. 6c). The
temporal variation of $\delta_2$ shows fluctuations from uni- to multimodal rainfall distributions along the flow distance (Fig. 6d). In
Moixent town, the automatic stream-gauge recorded a peak discharge of 630.6 m$^3$s$^{-1}$ at 08:20 UTC on 12 September (Table 1;
Figs. 2 and 8a).

During phase 2, a prominent and organized convective band moved from northeast to southwest of the domain, mainly
impacting the *Ramblas Salada, del Albujón, de Benipila* and de *Canalejas* from 10 to 19 UTC on September, 12 (Figs. 4b–c).
During its passage, the most affected basins were the *Ramblas Salada* and *de Canalejas*. Maximum hourly basin-area average
rainfall accumulation was close to 40 mm over the former basin at 14 UTC, with a fractional coverage close to 80%. The





kinematics of the convective structure over the *Rambla Salada* is clearly described by the strong change in sign of $\delta_1$, indicating

a clear up- to downstream movement of the heavy rain while featuring an acute unimodal distribution (Figs. 6e-h). Regarding

the *Rambla de Canalejas*, the passage of the convective structure brought hourly catchment-area average rainfall

accumulations ≥ 50 mm between 17–19 UTC, with fractional basin coverages above 70% (Figs. 6i-j). Again, the spatial

moments describe the motion of the linear precipitation structure, with a fast downstream translation and a marked unimodal

distribution (Figs. 5k-l). The burst of heavy precipitation led to an extraordinary flash flooding: two almost consecutive

maximum discharges of 367.4 $m^3s^{-1}$ and 475.0 $m^3s^{-1}$ were recorded in El Pinar town at 18:40 and 19:40 UTC on 12 September,

respectively (Fig. 8e).

During phase 3, intense and long-lasting rainfall affected the *Ramblas Salada, de Benipila, de Albujón and Almanzora* basins

from 19 UTC on 12 September to 06 UTC on 13 September (Figs. 4d–f). The *Rambla Salada* was partially impacted by hourly

basin-area average rainfall accumulations up to 20 mm (Figs. 6e–f). First, rains mainly affected the headwaters. Subsequent

convectively-driven precipitation impacted the catchment following an upstream direction. The rainfall distributions were

mostly unimodal (Fig. 6g–h). As a result of the intense and copious precipitation during phases 2 and 3, two successive

maximum flows of 107.8 and 105.4 $m^3s^{-1}$ were observed at 06:20 and 07:30 UTC on 13 September (Table 1; Figs. 2 and 8b).

The *Ramblas del Albujón* and *de Benipila* were first partially affected by the passage of the linear structure of precipitation

generated during phase 2 (12-16 UTC on September, 12), which resulted in maximum hourly basin-area average rainfall

accumulations ≥ 10 mm, mainly distributed over the headwaters for both basins (Figs. 7a–b and e–f). Scattered and

convectively-driven precipitation impacted the watersheds during the next 3 h, with a maximum cumulative precipitation above

15 mm over the *Rambla del Albujón* at 19 UTC. Finally, the MCS overwhelmed both catchments from 21 UTC onwards.

During its mature stage, hourly basin-area average rains were above 20 and 25 mm in the *Ramblas del Albujón* and *de Benipila*,

respectively. The catchment percentages covered by heavy precipitation during this 9-h period were of up to 50 and 70% of

the basin sizes, respectively. The quasi-stationary MCSs mostly impacted the lower areas of the *Rambla del Albujón*, with an

accentuated multimodal distribution (Figs. 7c-d). Regarding the *Rambla de Benipila*, the spatial moments indicate slight up-

and downstream oscillations of the unimodal spatial distributions of the intense downpours during the passage of the MCS

(Figs. 4e and 7g-h). The observed peak discharges were of 276.1 $m^3s^{-1}$ at 04:50 UTC in the *Rambla de Benipila*, and of 105.0

$m^3s^{-1}$ at 05:30 UTC over the *Rambla del Albujón* (Figs. 8c-d).

The passage of the linear convective structure across the *Almanzora* catchment brought copious precipitation from 19 to 23

UTC on 12 September, with hourly basin-area average rainfall accumulations about 20 mm and a maximum spatial coverage

above one third of the basin extension (Figs. 7i-j). Finally, the triggering of intense convective bands in western Almería that

moved northwest, impacted the *Almanzora* basin from 00 UTC on 13 September. Convective activity persisted over the

catchment during the next 7 hours (Figs. 4d–f). The lifting associated with topographic forcing enhanced rainfall efficiency,

giving rise to quasi-stationary precipitation: The maximum hourly basin-area average rain was above 30 mm at 01 UTC, with

a fractional basin coverage close to 50%. $\delta_1$ reflects the successive passages of the different convective systems, while $\delta_2$



denotes marked unimodal rainfall distributions (Figs. 7k-l). A maximum peak discharge of 283.2 m³s⁻¹ was recorded in Cantoria town at 04:00 UTC on 13 September (Fig. 8f).

## 4 Analyses of flood response

### 4.1 Flood response and catchment dynamics

The total basin-average cumulative precipitation fluctuated roughly from 135 to 226 mm, while runoff ratios were remarkably small, varying from 0.04 to 0.16 (except for the *Rambla Salada*; Table 1). The runoff coefficients do not exhibit any clear relationship with maximum hourly rainfall rate or total accumulation across the examined basins. These highly nonlinear hydrological responses can be ascribed to the combination of very dry initial soil conditions and large soil water capacities. Indeed, runoff deficits were especially severe on these basins with underlying fractured and karstified bedrock (i.e., the *Cànyoles* and *Almanzora* catchments) or with soils strongly altered by anthropogenic pressures (i.e., the *Rambla del Albujón*). As a result, specific peak discharges were uneven and rather small at basin scale, being only notable on the *Ramblas de Benipila* and *de Canalejas*.

The catchment response time is linked to basin size, runoff generation and hillslope and channel network routing. In turn, runoff triggering is intimately connected to the spatial and temporal distribution of the rainfall patterns. Herein, lag time is used as a proxy to characterize basin dynamics (Creutin et al. 2009). It is defined as the temporal difference between the centres of mass of the rainfall hyetograph and hydrograph, measuring the catchment response time from the beginning of precipitation. Besides acute heterogeneities in the rainfall-runoff transformation, pronounced nonlinearities also arose in the hydraulics of the selected basins. Lag times were little sensitive to drainage area, being more dependent on features in rainfall (Figs. 9a–d). This behaviour would reflect the composite effects of several factors: an increased amount of sheet flows on the hillslopes with cumulative precipitation, an expansion of the stream networks to previously unchanneled topographic elements, as well as an increase of flow velocity with discharge (Borga et al. 2007).

According to the lag time as function of basin size, three different modes of hydrological response were present during the unfolding of the 12-13 September 2019 widespread flash flooding (Figs. 9a). Despite the initial high abstractions, the *Rambla de Canalejas* and *Cànyoles* watersheds exhibited fast hydrological responses, with lag times of 2.7 and 3.5 h, respectively. For the latter catchment, features in precipitation did not stand out in comparison to the *Almanzora* basin, with similar size and physiography (Figs. 9b–d). However, heavy rainfall impacted the *Cànyoles* watershed during seven consecutive hours, being mostly focused on its lower part, according to the temporal average of δ1 over the most intense period (i.e., 00–12 UTC on September, 12; Figs. 6c–d and 9e–f). In the end, the rapid response of this basin can be mainly attributed to the particular spatial and temporal distributions of the rainfall patterns. The *Rambla de Canalejas* offered an exemplary case of fast infiltration-excess runoff generation to extreme precipitation rate: Rainfall was concentrated in just 3 h, leading to a total catchment-average amount of 147.6 mm, with relative basin coverages above 70% and a strong unimodal distribution along the flow path (Figs. 6i–j and 9b–f).




After the end of phase 2, the catchment-average total rains were of 44.4 mm in the *Rambla de Benipila*, 61.6 mm over the *Rambla de Albujón*, and 37.5 mm in the *Almanzora* watershed, but no appreciable runoff had yet been produced. Subsequent bursts of heavy precipitation during phase 3 led to fast infiltration-excess production, resulting in widespread flooding on these watersheds. Their hydrological responses were relatively slower than these observed in the *Cànyoles* and *Rambla de Canalejas* catchments, with lag times ranging from almost 5 to 8 h (Table 1; Figs. 9a). The delay in runoff activation points to cumulative

precipitation as another essential factor for flood control during this episode: Until the runoff thresholds were not surpassed, sudden hydrological response did not start. It is likely that total rainfall amount was also the main ingredient triggering sudden Hortonian runoff generation on the *Cànyoles* basin.

    The *Rambla Salada* basin had a very different hydrological response owing to its particular physiography. The total basin-average rainfall estimates were of 31.4, 77.5 and 116.7 mm during phases 1, 2 and 3, respectively. In spite of featuring the

largest amount of precipitated water, the second largest hourly rainfall rate and basin coverage, as well as the highest runoff coefficient, lag time was of 14.5 h (Table 1; Fig. 9). The rising and recession limbs of the observed hydrograph were more gradual (Fig. 8). The catchment response departed from the paroxysmal runoff processes and overland and channel flows typical of flash floods, as subsurface processes modulated flux dynamics (Table 1; Fig. 9). Owing to the important contribution of subsurface flow, it took around 8 h for the flow to recede to 30% of the second peak discharge. The exponential decay of

the remaining recession limbs was much faster. As an illustrative example, it only took 1.2 h for the observed hydrographs to recede in the same proportion over the *Cànyoles* and *Rambla de Canalejas* watersheds, exemplifying the predominant contribution of fast overland flow during the 12-13 September 2019 episode.

### 4.2 Hydrological modelling

    Hydrologic response is further analysed by using the event-based and fully-distributed Kinematic Local Excess Model (KLEM;

Da Ros and Borga 1997). This hydrological model accounts for properties in topography, soil and vegetation. Runoff rate ($q(x,y,t)$) at a given location and time (x, y, t), is computed from precipitation rate ($P(x,y,t)$) by employing the Soil Conservation Service-Curve Number method (CN; USDA 1986). The identification of the drainage network requires the characterization of hillslope and channelled paths by means of a threshold area procedure for catchment channelization. Next, the description of drainage system response is used to represent runoff routing (Giannoni et al. 2003). Discharge ($Q(t)$) at any

location along the stream is calculated as:

$$Q(t) = \iint_A q[x, y, t - \tau(x,y)]dxdy , \qquad (5)$$

where A represents the drainage area to the specific outlet location, and τ(x, y) is the routing time from point (x, y) to the outlet of the catchment. τ(x, y) is defined as:

$$\tau(x,y) = \frac{L_h(x,y)}{v_h} + \frac{L_c(x,y)}{v_c}, \qquad (6)$$

where $L_h(x,y)$ stands for the distance from the generic basin location (x, y) to the channel network following the steepest descent path; and $L_c(x, y$ is the length of the subsequent channel path to basin outlet. That is, surface and channel flow routing


depend on two constant velocities along the hillslopes ($v_h$) and channels ($v_c$) of the drainage network. KLEM models baseflow by using a linear conceptual reservoir based on the Horton-Izzard equation (Moore and Bell 2002).

Landscape morphologies and soil properties are described by a 25-m grid size cell. Curve numbers are derived from lithology

and land use maps, set to dry antecedent moisture conditions (i.e., AMC I) and kept constant during the calibration process (Table 2). The hydrological model is driven by the 10-min radar-derived QPEs from 18 UTC on 11 September to 00 UTC on 15 September. The computational model time-step is the same that the 10-min radar observing frequency.

Calibration tasks addresses peak discharge, time to peak and runoff volume, which are mainly modulated by the infiltration and the hillslope and channel flow velocities. The CN model is considered as a suitable conceptual scheme to describe the

runoff generating processes associated to the 12-13 September 2019 episode. Recall that direct runoff volume ($V_r(t)$) only depends on cumulative precipitation ($P(t)$) at a specific time from the beginning of the storm (t; Maidment 1993):

$$V_r(t) = \begin{cases} \frac{(P(t)-I_a)^2}{(P(t)-I_a+S)} & if\ P(t) > I_a \\ 0 & if\ P(t) \le I_a \end{cases}, \tag{7}$$

where $I_a$ denotes the runoff threshold and $S$ represents the soil retention capacity. $S$ is a site storage parameter described by the curve number, while $I_a$ is defined as a fraction of $S$ (Ponce and Hawkins 1996):

$$S = S_0 \cdot \left(\frac{100}{CN} - 1\right), \tag{8}$$

$$I_a = \lambda \cdot S, \tag{9}$$

where $S_0$ stands for the infiltration storativity and $\lambda$ is the initial abstraction ratio. $S_0$ and $\lambda$ are considered as adjustment parameters so as to better deal with the initial very dry soil conditions and large storage capacities. Calibration of $S_0$ and $\lambda$ enables to simulate correctly the observed water balance (Borga et al. 2007). Heterogeneities in hydraulics have been

encompassed by calibrating the hillslope and channel flow velocities (Table 2).

**4.3 Model performance and water balance**

The performance of QPEs driven runoff simulations is evaluated by means of the Nash–Sutcliffe efficiency criterion (NSE; Nash and Sutcliffe 1970) and LNP cost function (Roux et al. 2011). The skill of the hydrological experiments is also analysed in terms of the relative errors in peak discharge and total direct runoff volume, expressed as a percentage. The calibrated

KLEM simulations succeed when performing the different hydrological responses (Tables 2 and 3, and Fig. 8). The remarkably high NSE and LNP cost function scores indicate a good general reproduction of the observed hydrographs in terms of peak discharge and timing. However, the observed water balances are considerably overestimated in the *Cànyoles*, *Rambla de Benipila* and *Rambla de Canalejas* catchments. The overall successful performance of the numerical experiments can be partially ascribed to the strong role of the heavy rainfall when modulating flood response for the case under study, as QPEs

have been estimated with a relatively good accuracy over the region of interest (Figs. 3 and 4). It seems unlikely that imprecisions in the simulated water balances emerge from large errors in rainfall estimates over the concerned watersheds. Automatic rain-gauge density is relatively high on and around the catchments, minimizing possible biases. These inaccuracies



would be more attributable to errors in the precise description of soil and geological properties, which are always difficult to assess.

Imprecisions in the simulated water balances mainly arise due to inaccuracies in the reproduction of the observed hydrograph limbs (Table 3 and Fig. 8). Errors in the simulated recession branches for the *Cànyoles* and *Rambla de Canalejas* basins would denote that soil did not reach saturation. Abstractions remained despite the large volume of water previously precipitated on both watersheds. For the former watershed, these losses may be ascribed to the underlying karst geology over large portions of the basin. Imprecisions also emerge when simulating the observed rising limb in the *Rambla de Benipila*, resulting in an

excessive initial runoff volume and a double peak discharge (Fig. 8d). This watershed combined high initial abstractions with a sudden subsequent overflow production, resulting in a remarkable peak discharge once runoff thresholds were exceeded. It is also arduous to accurately reproduce the fast processes involved in increasing the efficiency of the rainfall-runoff transformation and flow routing, while notable abstractions still remain. According to the observed hydrograph in the *Rambla de Canalejas*, a sharp decrease in catchment response to increased rainfall amount occurred: the main peak discharge was just

1 h apart of the previous relative maximum, being quite sudden and narrow (Fig. 8e).

With the exception of the *Rambla Salada*, the simulated overland flow speeds vary from 0.18 to 0.35 ms$^{-1}$, whereas channel flow velocities range from 3.5 to 4.0 ms$^{-1}$ (Table 2). The overland flow velocities are high as a result of the sparse vegetation and large amounts of sheet flow produced on the hillslopes during flash flooding. Since KLEM assumes the overland flow to be constant in space and time, high overland flow speeds allow to account for sheet flow as well as concentrated overland flow

in not previously channelized areas (Borga el at. 2007). Channel flow velocities are also high, as a consequence of the lack of vegetation in ephemeral stream beds and on banks, relatively steep slopes and increase of flow velocity with discharge.

## 5 Scale dependency between flood magnitude and rainfall amount

### 5.1 Analytical framework

A fundamental research topic in hydrometeorology is to delve into the physical basis of observed self-similarity linking peak discharge to other physical variables. As runoff and flow routing processes are scale-dependent on cumulative precipitation in extreme flash flooding, the connection between flood magnitude and total rainfall amount is further investigated through scale-invariance. Schumer et al. (2014) observed that a power-law relationship links peak discharge ($Q_p [m^3/s]$) to total flow volume

($V_r [m^3]$) for a given episode duration in arid and semi-arid basins dominated by Hortonian runoff generation:

$$V_r = a \cdot Q_p^b, \qquad (10)$$

where the regression coefficient ($a [s^b/m^{3(b-1)}]$)) is the reduced discharge volume; and $b$ is the flood-scaling exponent. Direct runoff volume and cumulative precipitation for the same episode and duration are also related as:

$$V_r = \frac{1}{3.6} A \cdot C_r \cdot P, \qquad (11)$$




where $A$ $[km^2]$ is the drainage basin, $C_r$ $[-]$ is the event runoff coefficient, $P$ $[mm]$ is the total catchment-average rainfall

amount and $1/3.6$ is for unit conversion. By combining equations (10) and (11), specific peak discharge ($q_p$ $[m^3/(s \cdot km^2)]$)

is related to cumulative precipitation by the following scaling-law:

$$q_p = A^{\frac{1-b}{b}} \cdot \left(\frac{C_r}{3.6 \cdot a}\right)^{1/b} \cdot P^{1/b} \equiv \alpha \cdot P^{\beta}, \tag{12}$$

where the regression coefficient ($\alpha$ $\left[m^3/\left(s \cdot km^2 \cdot mm^{\beta}\right)\right]$) is the reduced specific peak discharge; and $\beta$ is the flood-scaling

exponent. Expressions (10) and (12) exhibit an inverse relationship between the algebraic exponents of both power-laws.

Schumer et al. (2014) also indicated that parameters a and b are related to catchment and rainfall properties. This assumption

can be readily checked by considering the limiting case of a linear dependence between flood magnitude and total precipitation

(i.e., $\beta = 1$). In this particular limit, equations (10) and (12) become:

$$V_r = a \cdot Q_p, \tag{13}$$

$$q_p = \frac{1}{3.6 \cdot a} \cdot C_r \cdot P, \tag{14}$$

where $a$ in eq. (13) matches the definition of flood timescale ($T_Q$ $[s]$) introduced by Gaál et al. (2012):

$$T_Q = \frac{V_r}{Q_p}, \tag{15}$$

Therefore, equation (14) can be expressed as follows:

$$q_p = \frac{1}{3.6 \cdot T_Q} \cdot C_r \cdot I \cdot T_m = \frac{1}{3.6 \cdot \sqrt{Var(t_Q)}} \cdot R \cdot T_m, \tag{16}$$

where $I$ $[mm/s]$ is the storm- and basin-average rainfall rate, and $T_m$ $[s]$ denotes the storm duration. $R$ $[mm/s]$ stands for the

storm- and catchment-average excess rainfall, $t_Q$ $[s]$ corresponds to the timing of runoff (i.e., the characteristic time over which

runoff is distributed), and $Var(t_Q)$ is the temporal dispersion of the runoff hydrograph. The inverse of the temporal dispersion

of the runoff hydrograph measures the peakedness of the hydrograph. Viglione et al. (2010) found that an almost mathematical

equality stands between the flood timescale and standard deviation of timing in runoff. Equation (16) is the response number,

a measure of flood peak magnitude also defined by Viglione et al. (2010). In this limiting case, the scaling relationship becomes

the response number, that integrates the joint effect of runoff coefficient and hydrograph peakedness in determining flood

peak.

In the general case, equation (12) assesses the nonlinear interaction between storm properties and hydrologic processes for

particular arid or semi-arid catchment specificities by only considering two integrated variables: specific peak discharge and

cumulative precipitation. Regression coefficients and flood-scaling exponents differing from unity quantify the degree at which

basins filter spatial and temporal variability in rainfall. On the one hand, as an increased precipitation amount results in an

enhanced efficiency of the rainfall-runoff processes, basins are expected to have flood-scaling exponents greater than 1. On

the other hand, heterogeneities in soil and underlying substrate anticipate regression coefficients much smaller than unity. The

smaller the reduced specific peak discharge, the larger the impact of these features on the nonlinear hydrological response.





Finally, the power law relationship also allows to examine how precipitation amount modulates flood magnitude for different return periods.

## 5.2 Numerical exploration

The 12-13 September 2019 episode offers a suitable benchmark to further investigate the general features in the scale
dependency between flood peak and rainfall amount. The unusual spatial and temporal extension of the responsible HPE, together with the contrasting physical features of the examined watersheds, permits cross-validations and inter-comparisons. Amengual et al. (2021) devised a set of distinct ensemble prediction systems (EPSs) based on a convection-permitting numerical weather prediction model in order to examine predictability of the 12 and 13 September 2019 HPE at regional and catchment scale. These ensemble strategies also provide valuable information about plausible and equally-likely rainfall
scenarios for this particular episode. Therefore, it is possible to examine how each basin integrates the diverse collection of rainfall intensities and amounts. Consequently, all the QPFs driven runoff simulations are used to further examine the scaling properties linking peak discharge and total rainfall amount.

In the end, five different 50-member EPSs were generated for two consecutive 24-h periods aiming to cope with all meteorological uncertainties. Next, the quantitative precipitation forecasts (QPFs) were used to force the KLEM model. Note
that this procedure considered up to 500 different rainfall scenarios (i.e., 5 EPSs x 50 ensemble members each x 2 initialization days) impacting on each individual watershed. Again, regression coefficients and flood-scaling exponents are calculated by using OLS regression; uncertainty is quantified by applying a 1000-sample bootstrap with replacement, while statistical significance is checked by means of the p-value. Certainly, although this numerical exploration is exclusively limited to the fixed set of hydrological model parameters describing each basin response for the 12-13 September 2019 episode (Table 2), it
can be considered as a valuable experiment so as to provide some guidance in relation with the different basin responses.

The *Rambla de Canalejas* and *Almanzora* watersheds feature the most efficient hydrological responses to large cumulative precipitation (Table 4 and Fig. 10). As expected, the most effective catchments are the steepest (Fig. 2), as they promote a fast generation of runoff, sheet flow and concentrated overland flow, as well as a quick flow channel routing. Despite being the most effective, these catchments evidence the largest nonlinear rainfall-runoff transformations: besides having the highest
flood-scaling exponents, also exhibit very low regression coefficients. The acute nonlinearity of the *Almanzora* response may be ascribed to the dominant presence of highly karstified and fractured bedrock. The efficient rainfall-runoff conversion for the *Rambla de Canalejas* is probably related to the fast processes involved in increasing runoff generation and propagation with enhanced cumulative precipitation. Nevertheless, it also features the weakest squared correlation coefficient, as a high variability is found for the smallest specific peak discharges in terms of rainfall amount (Table 4, Fig. 10e).

The *Ramblas del Albujón* and *de Benipila* feature a smaller enhancement in the effectiveness of the runoff and routing processes at increased precipitation, partially because both watersheds are flatter (Table 4; Fig. 2). In addition, the *Rambla del Albujón* has a comparatively small scaling coefficient, probably because it is a highly human-modified basin. The hydrological response over the *Cànyoles* basin is halfway to the behaviour of the previous watersheds. Besides having comparable mean slopes, this


outcome may also be attributed to the fact that rainfall amounts are not so extreme over this catchment. Indeed, only one
rainfall scenario renders cumulative precipitation above the 10-yr return period (Fig. 10a).

The *Rambla de Benipila* yields more substantial peak discharges for a particular rainfall amount than the remaining basins.
Specific peak discharges are always above 0.02 $m^3s^{-1}km^{-2}$, independently of the cumulative precipitation (Fig. 10d). This
output reflects the aforementioned difficulties when simulating high initial abstractions and sudden subsequent runoff
productions in this watershed. The *Rambla Salada* departs from the paroxysmal dynamics of arid and semi-arid basins under
flash flood conditions owing to the dominance of subsurface processes (Tables 1 and 2). Accordingly, this watershed does not
follow a scaling-flood law with cumulative precipitation, as its internal hydrological processes result in less efficient rainfall-
runoff conversion and flow propagation at enlarged rainfall amount (Fig. 10b). When subsurface flow processes are dominant,
the scaling-law no longer remains.

**6 Conclusions and further remarks**

In line with previously published flash-flood monographs, this work analyses the main hydrometeorological characteristics of
the 12-13 September 2019 extreme HPE that impacted the central and southern part of Mediterranean Spain. This episode
offers a valuable benchmark to examine the main rainfall and runoff processes in the context of similar observational studies
of catastrophic flash flooding in the Western Mediterranean region, within the heavy rainfall, flash-floods and floods section
of HyMeX.

The HPE was divided into three stages from an evaluation of the different convective structures: phase 1 consisted of a quasi-
stationary and elongated area of anchored convection, resulting in long-lasting precipitation over the northernmost watershed;
phase 2 was the result of a linear convective system impacting the central and southern basins, and; in phase 3, the formation
and subsequent evolution of a V-shaped MCS overwhelmed the central catchments, while intense convective band activity
was affecting the southernmost basin. One of the most striking characteristics of this HPE was the large spatial extent of rainfall
accumulations ≥ 200 mm. According to the QPEs, while the areas of cumulative precipitation equal or larger than 400 mm
were very limited, cumulative precipitation ≥ 200 mm extended up to 7500 $km^2$. This vast spatial extension resulted in
fractional basin coverages ranging from 0.5 to 0.9 during the most intense rainfall periods. Besides quasi-stationarity, another
striking feature was the acute and persistent convective activity during the unfolding of this episode. The different spatial and
temporal scales of convectively-driven rainfall resulted in a self-similar organization between maximum 30-min rainfall rate
and cumulative precipitation.

Despite the impressive spatial and temporal extension of the heavy rains, very low antecedent soil moisture contents together
with large infiltrabilities resulted in a remarkable damping of the flood response. The strong role of the initial conditions was
another noticeable characteristic of the 12-13 September 2019 hydrometeorological episode: Runoff coefficients ranged
between 0.04–0.16 over the flashiest basins, resulting in highly nonlinear hydrological responses. Besides precipitation rate,
rainfall amount arose as another major control in flood magnitude and dynamics over most catchments. Until runoff thresholds



were not exceeded, no sudden infiltration-excess runoff generation started. This flood-triggering process resulted in a delayed hydrological response, but once runoff started, the rainfall-runoff and dynamical processes became more efficient at increased rainfall amount.

As cumulative precipitation can also be an important factor when modulating flood magnitude, how rainfall amount translates into scale-dependent peak discharge was further investigated through simple scaling theory. A power-law relationship was analytically derived from previous research in arid and semi-arid basins controlled by surface flow dynamics under flash flood conditions. Subsequent numerical exploration confirmed the statistical robustness of the adjusted power-law relationships for the selected catchments under a particular set of parameterizations. Intercomparisons suggest that the flood-scaling coefficient

and exponent may be mostly dependent on the spatial and temporal variability in soil physiography and rainfall, as well as on basin morphology. Admittedly, hydrological response linked to the exceedance of runoff thresholds may have important implications for estimating flood frequencies in arid and semi-arid basins with limited observed flow records. As alternative, the proposed scaling relationship could serve as basis to obtain empirically-derived envelope curves.

Recognizably, this work is just a preliminary attempt to further delve into self-similarity between flood magnitude and

cumulative precipitation over arid and semi-arid Hortonian basins, as it only relies on this particular case study. The intercept and flood scaling parameters are expected to be case-dependent. Future research based on long-term, high-resolution observations must establish how this relationship varies for a larger sample of extreme episodes in terms of properties in rainfall (i.e., intensity and type), catchment (i.e., physiography and morphology), and initial soil moisture conditions (i.e., seasonality). The successful fulfilment of these tasks would allow to derive suitable envelope curves to better appraise the

upper bound of flood events depending on total rainfall amount. For instance, these empirically-derived envelope curves may be useful for hydrological design purposes and flood risk management. In this sense, Ewea et al. (2020) has recently shed some light on these issues after analyzing several envelope curves derived from long-term daily series of maximum flood records in the arid environment of Saudi Arabia.

**Acknowledgements**

A. Amengual is grateful to M. Borga for his helpful comments during the elaboration of this work. This study was sponsored by the FEDER/Ministerio de Ciencia, Innovación y Universidades - Agencia Estatal de Investigación/COASTEPS (CGL2017-82868-R) and TRAMPAS (PID2020-113036RB-I00/AEI/10.13039/501100011033) research projects. The Confederaciones Hidrográficas del Júcar y del Segura, la Demarcación Hidrográfica de las Cuencas Mediterráneas Andaluzas and the Spanish

Agency of Meteorology are acknowledged for providing the data needed to perform this work.



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



# TABLES

| Basin | Gauge | Total rainfall (mm) | Total runoff (mm) | Peak discharge ($m^3s^{-1}$) | Specific peak discharge ($m^3s^{-1}km^{-2}$) | Runoff ratio (–) | Lag time (h) |
|-------|-------|------|------|------|------|------|------|
| Cànyoles | Moixent | 134.8 | 7.3 | 630.6 | 0.72 | 0.05 | 3.5 |
| Salada | Salada | 225.6 | 62.5 | 107.8 | 0.97 | 0.28 | 14.5 |
| Albujón | Albujón | 141.8 | 5.1 | 105.0 | 0.27 | 0.04 | 8.0 |
| Benipila | Benipila | 151.4 | 21.2 | 276.1 | 1.95 | 0.14 | 4.8 |
| Canalejas | El Pinar | 172.5 | 26.9 | 475.0 | 3.74 | 0.16 | 2.7 |
| Almanzora | Cantoria | 181.6 | 9.2 | 283.2 | 0.26 | 0.05 | 8.0 |

**Table 1 Main hydrometeorological features of the 12-13 September 2019 widespread flash flooding for the different hydrometric sections of the selected catchments. Total rainfall amounts are radar-derived and are expressed as area-average values over the basins. Lag time is computed as the temporal difference between the rainfall and runoff centers of mass.**

| Basin | Gauge | CN (AMC I) | $S_0$ (mm) | $\lambda$ | $V_h$ ($ms^{-1}$) | $V_c$ ($ms^{-1}$) |
|-------|-------|------------|-----------|-----------|-------------------|-------------------|
| Cànyoles | Moixent | 48.3 (10.6) | 355.6 | 0.40 | 0.35 | 3.5 |
| Salada | Salada | 48.8 (12.0) | 330.2 | 0.40 | 0.04 | 2.3 |
| Albujón | Albujón | 40.6 (10.9) | 342.9 | 0.50 | 0.30 | 3.8 |
| Benipila | Benipila | 42.3 (11.4) | 254.0 | 0.20 | 0.18 | 3.5 |
| Canalejas | El Pinar | 46.4 (12.4) | 279.4 | 0.30 | 0.35 | 3.5 |
| Almanzora | Cantoria | 48.3 (12.9) | 406.4 | 0.60 | 0.18 | 4.0 |

**Table 2 KLEM parameters for infiltration and dynamical processes and the different hydrometric sections of the selected catchments. Curve numbers are expressed as area-averaged values, while their standard deviations are shown in brackets. Note that curve numbers correspond to dry antecedent conditions and that the calibrated parameters were $S_0$, $\lambda$, $V_h$ and $V_c$.**





| Basin | Gauge | Flow volumes | | | Flow peaks | | | NSE | $L_{NP}$ |
|---|---|---|---|---|---|---|---|---|---|
| | | OBS (mm) | KLEM (mm) | Rel. error (%) | OBS ($m^3s^{-1}$) | KLEM ($m^3s^{-1}$) | Rel. error (%) | | |
| Cànyoles | Moixent | 7.3 | 9.6 | 32.8 | 630.6 | 604.7 | -4.1 | 0.77 | 0.89 |
| Salada | Salada | 62.5 | 62.5 | 0.0 | 107.8 | 107.9 | 0.1 | 0.91 | 0.89 |
| Albujón | Albujón | 5.1 | 5.6 | 10.4 | 105.0 | 102.2 | -2.7 | 0.91 | 0.89 |
| Benipila | Benipila | 21.2 | 28.9 | 36.0 | 276.1 | 265.5 | -3.8 | 0.77 | 0.90 |
| Canalejas | El Pinar | 26.9 | 35.6 | 32.6 | 475.0 | 479.2 | 0.9 | 0.76 | 0.84 |
| Almanzora | Cantoria | 9.2 | 8.6 | -6.9 | 283.2 | 286.9 | 1.3 | 0.94 | 0.97 |

**Table 3 Observed and radar-driven simulated flow volumes and peak discharges for the different hydrometric sections of the selected catchments. Model performance also shown in terms of the different skill scores. Negative values in relative errors denote model underestimation. NA denotes data not available.**


| Basin | Gauge | $\alpha$ ($m^3s^{-1}km^{-2}mm^{-\beta}$) | $\beta$ | $R^2$ |
|---|---|---|---|---|
| Cànyoles | Moixent | 2.5 (3.9–1.6)·$10^{-6}$ | 2.3 (2.2–2.4) | 0.84 |
| Salada | Salada | – | – | – |
| Albujón | Albujón | 4.9 (7.7–2.9)·$10^{-7}$ | 2.7 (2.6–2.9) | 0.91 |
| Benipila | Benipila | 7.1 (12.5–4.2)·$10^{-5}$ | 2.0 (1.9–2.1) | 0.93 |
| Canalejas | El Pinar | 1.3 (4.4–0.3)·$10^{-7}$ | 3.3 (3.0–3.7) | 0.80 |
| Almanzora | Cantoria | 3.1 (4.3–2.7)·$10^{-9}$ | 3.6 (3.5–3.6) | 0.92 |

**Table 4 Reduced discharges ($\alpha$), flood-scaling exponents ($\beta$), and squared correlation coefficients ($R^2$) for the selected basins at the indicated stream-gauges. Note that hydrological response does not follow a power-law relationship in the Rambla Salada. Also note that p-values are less than 0.01 for all the regressions. The 95% confidence intervals for $\alpha$ and $\beta$ are shown between parentheses.**




**FIGURES**


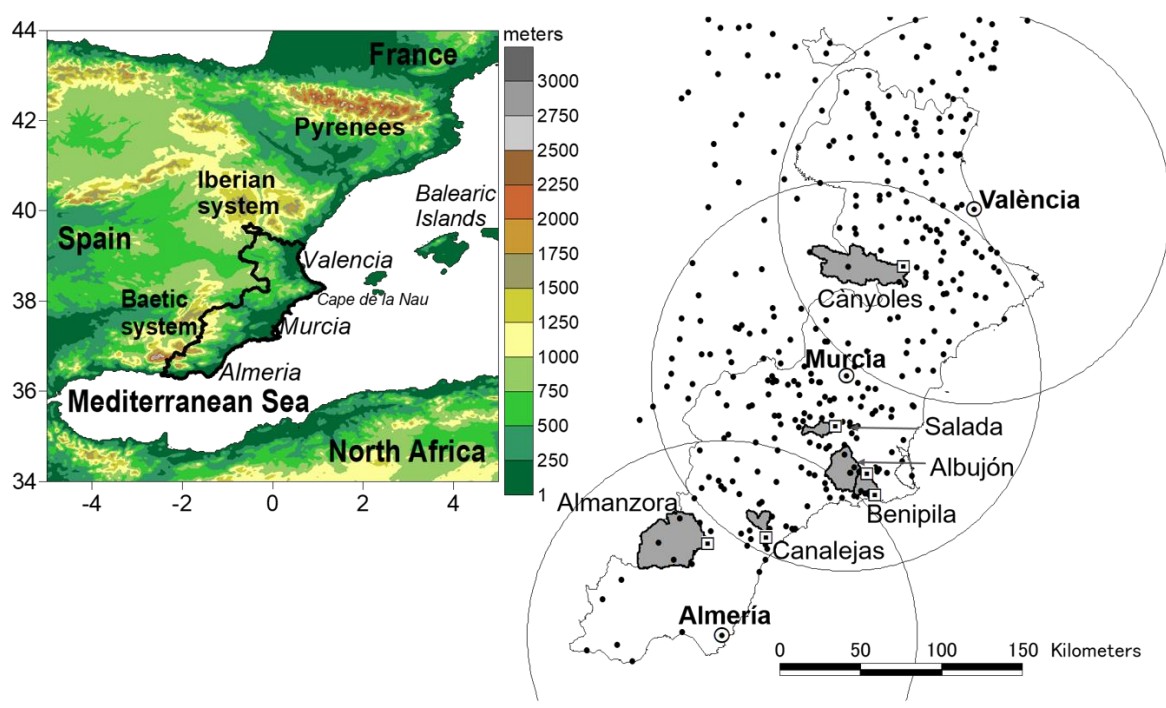

**Figure 1 Top left figure: Main geographical features of the Spanish Mediterranean region. The thick black line delineates the study area. Central right figure: Location of the València, Murcia, and Almería radars (white dots) and the selected catchments (light grey shaded areas). Radii of the radar circles are 120 km. The 369 available automatic pluviometric stations are shown as black**
**dots. Automatic stream-gauges are depicted as white squares.**





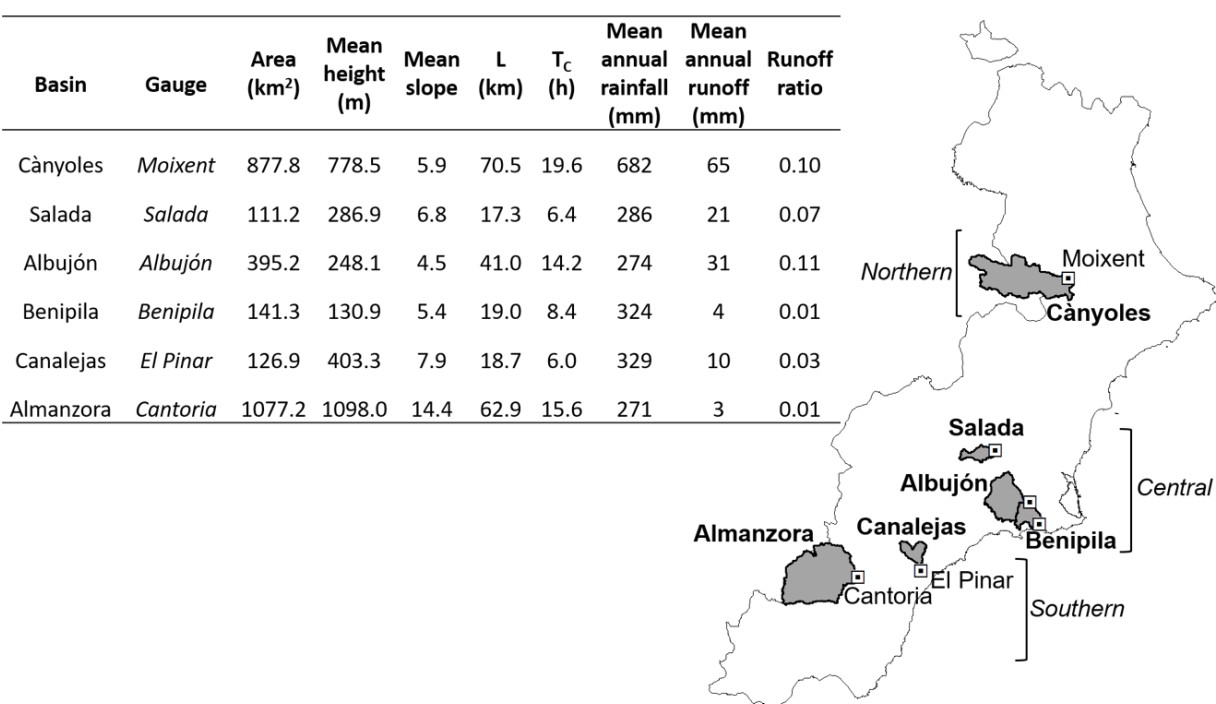

| Basin | Gauge | Area (km²) | Mean height (m) | Mean slope | L (km) | $T_c$ (h) | Mean annual rainfall (mm) | Mean annual runoff (mm) | Runoff ratio |
|---|---|---|---|---|---|---|---|---|---|
| Cànyoles | *Moixent* | 877.8 | 778.5 | 5.9 | 70.5 | 19.6 | 682 | 65 | 0.10 |
| Salada | *Salada* | 111.2 | 286.9 | 6.8 | 17.3 | 6.4 | 286 | 21 | 0.07 |
| Albujón | *Albujón* | 395.2 | 248.1 | 4.5 | 41.0 | 14.2 | 274 | 31 | 0.11 |
| Benipila | *Benipila* | 141.3 | 130.9 | 5.4 | 19.0 | 8.4 | 324 | 4 | 0.01 |
| Canalejas | *El Pinar* | 126.9 | 403.3 | 7.9 | 18.7 | 6.0 | 329 | 10 | 0.03 |
| Almanzora | *Cantoria* | 1077.2 | 1098.0 | 14.4 | 62.9 | 15.6 | 271 | 3 | 0.01 |

**Figure 2 Top left figure: Main physical features and climatological water balance of the catchments at the examined stream-gauges. L stands for the length of the main river channel. Tc is the time of concentration computed according to Amengual et al. (2021). Data for hydrologic balance come from the Spanish Ministry of Agriculture. Water balance was calculated for the 1941-2006 period. Further information at: https://www.miteco.gob.es/es/cartografia-y-sig/ide/descargas/agua/ simpa.aspx. Central right figure: The selected catchments are shown as light grey shaded areas. Automatic stream-gauges are depicted as white squares.**





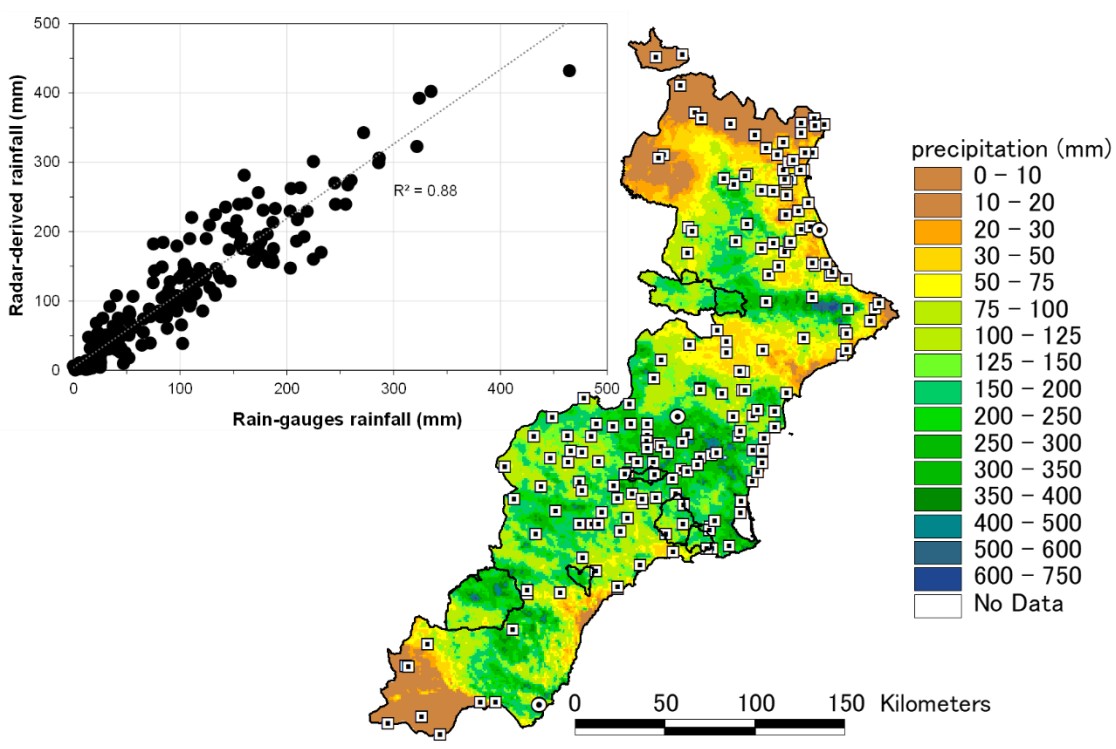

**Figure 3 Top left figure: scatterplot of the 48-h radar rainfall estimates against observed accumulations by the daily rain-gauge**
**network. Central right figure: spatial distribution of the 48-h accumulated radar-estimated precipitation from 12 to 14 September**
**2019 at 00 UTC over the study region. Thick black lines denote hydrographic basins. White squares stand for daily pluviometric**
**stations. White dots show the position of weather radars.**






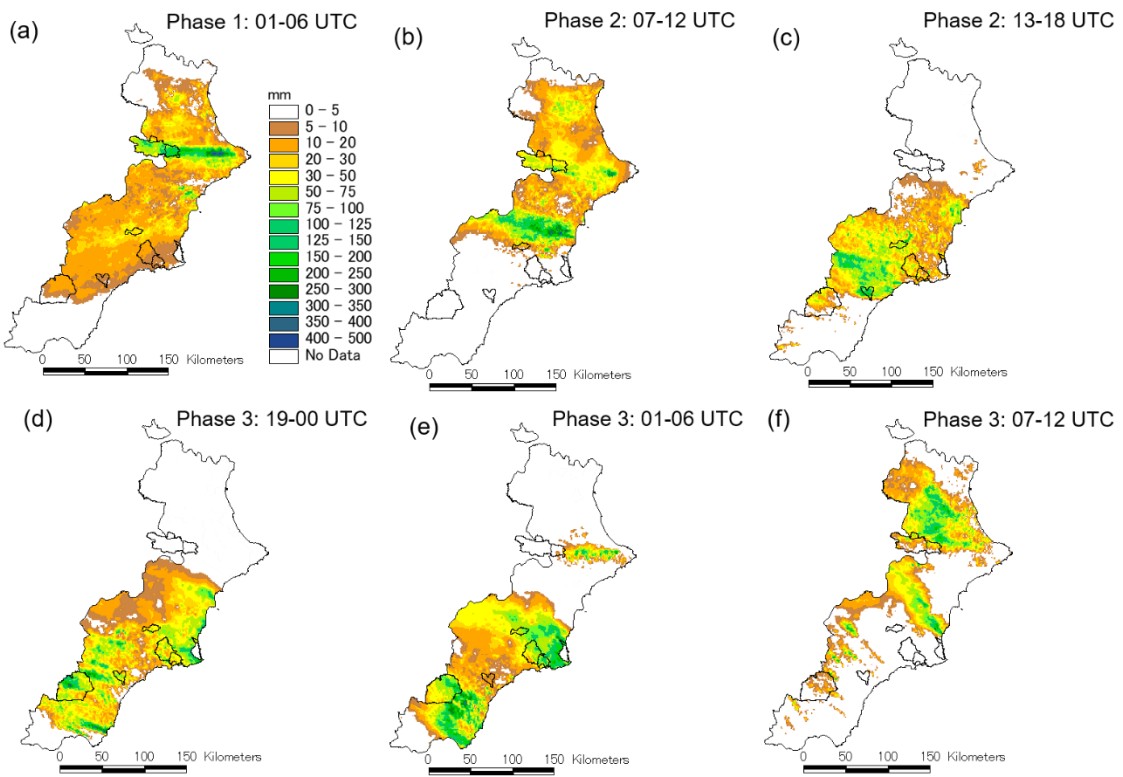

**Figure 4 Spatial distribution of 6-h cumulative radar-estimated rainfall corresponding to: (a) 01–06, (b) 07–12, and (c) 13–18 UTC on 12 Sep 2019; (d) from 19 UTC on 12 Sep 2019 to 00 UTC on 13 Sep 2019; and (e) 01–06 and (f) 07–12 UTC on 13 Sep 2019. Thick black lines denote the catchments of interest.**







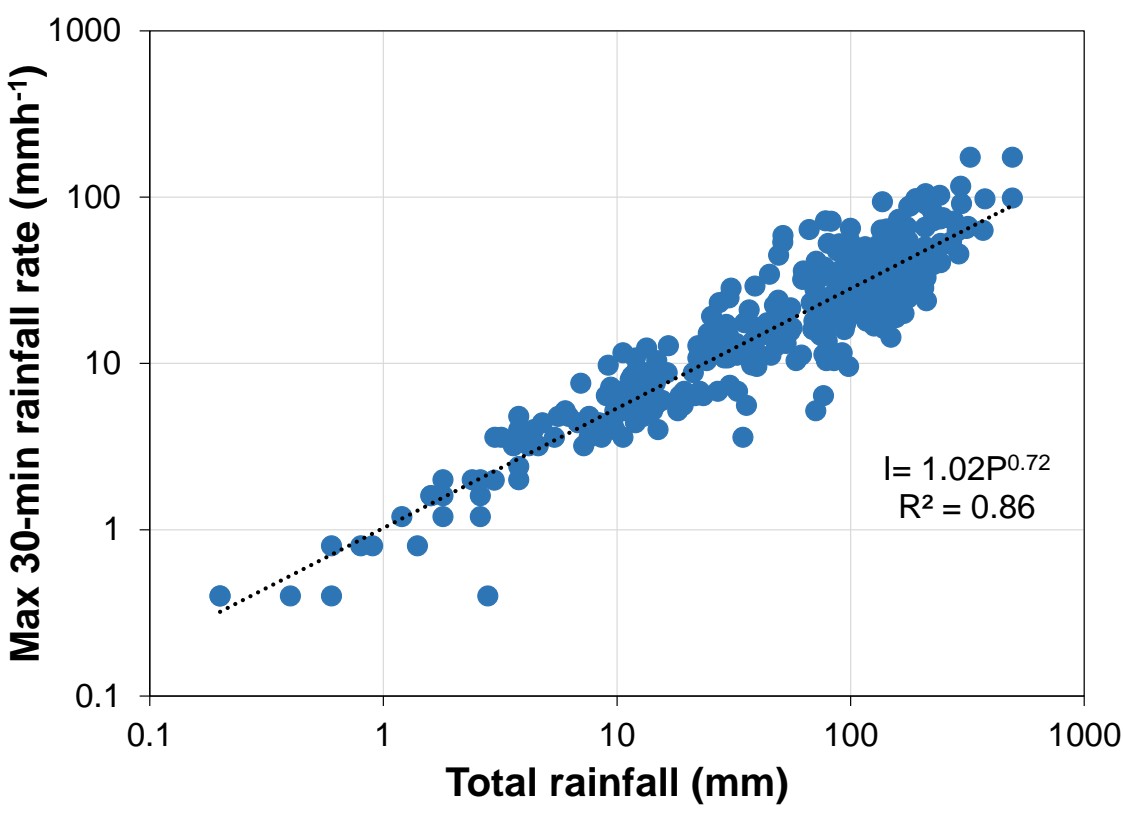

**Figure 5 Maximum 30-min rainfall rates versus total rainfall amounts from 12 to 14 September 2019 00 UTC for the 369 automatic rain-gauges located inside the area of interest.**







**Figure 6 Hourly time series over the most intense rainfall periods showing: the catchment-area average rainfall, fractional basin area covered by hourly precipitation > 20 mm, $\delta_1$ and $\delta_2$ for the Moixent (a–d), Rambla Salada (e–h) and Rambla de Canalejas (i–l) watersheds.**





**Figure 7 As in Fig. 6, but for the Rambla del Albujón (a–d), Rambla de Benipila (e–h) and Almanzora (i–l) watersheds.**




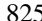

**Figure 8 Observed and KLEM radar-driven discharge simulations for the 12-13 September widespread flash flooding at: (a)
Moixent, (b) Salada, (c) Albujón, (d) Benipila, (e) Pinar, and (f) Cantoria flow-gauges. Also shown are the 10-min area-average
precipitations enclosed by the different hydrometric sections. Red dashed lines indicate 10-yr return periods for peak discharges**
**where available (labelled as Q10).**






**Figure 9 Lag time with respect to the: (a) basin size, (b) total catchment-area average precipitation, (c) maximum hourly rainfall rate, (d) maximum fractional basin coverage, and (e, f) first and second temporal-average spatial moments over the most intense rainfall period.**




**Figure 10 Specific peak discharges (qp) versus total rainfall amounts (P) for radar-derived (black rhombus) and ensemble experiments (blue dots) at the: (a) Moixent, (b) Salada, (c) Albujón, (d) Benipila, (e) Pinar, and (f) Cantoria hydrometric sections. Red dashed lines denote specific peak flow and total rainfall quantiles for a 10-year return period.**