# Peer review of "Hydrometeorological analysis of the 12 and 13 September 2019 widespread flash flooding in eastern Spain"

_Natural Hazards and Earth System Sciences, 2021_

## Author Response (AR1)

**Universitat de les Illes Balears**
Arnau Amengual Pou
Grup de Meteorologia
Departament de Física
Ctra. Valldemossa km 7.5
07122 Palma, Mallorca, Spain
arnau.amengual@uib.es

Palma, 08 February 2022

Dr. Christian Barthlott
Natural Hazards and Earth System Sciences

Dear Dr. Barthlott,

A revised version of the manuscript nhess-2021-386 entitled: "Hydrometeorological analysis of the 12 and 13 September 2019 widespread flash flooding in eastern Spain" by A. Amengual is attached. The author has addressed all comments and concerns pointed out by the reviewers. In next pages, the author includes a point-by-point response to their specific comments and concerns.

Sincerely,

Dr. Arnau Amengual

**REVIEWER 1**

**General comments**

**Author presents an interesting paper very dense of meteorological and hydrological information. In a sense I would say too dense. Author describes the 12 and 13 September 2019 very accurately, providing analysis of precipitation, even with the help of spatial moments of catchment rainfall, and analysis of flood response including runoff coefficients and lag times. Then a reconstruction of the model through a spatial distributed hydrological model is provided, including model performance analysis. Then author introduces a scale dependency analysis framework to relate rainfall to discharge. Finally the framework is applied by using data from an ensemble prediction system.**

**With all these points presented, I cannot understand the real objective of the work. From the title I understand this paper should be an analysis of the flood event, but more than this is presented. Author himself states that there are two objectives (L 90-97). I understand that the final objective should be to evaluate methods to assess hydrological risk in semiarid basins, but I think results presented are at too early stage.**

**So, given that the topic of the paper is very interesting to NHESS readers, I suggest to simplify the paper by removing section 5, not because it is not relevant, but to give it the proper space in a dedicated paper, to be submitted when work will be more mature.**

The author would like to thank the highly constructive comments and suggestions made by the reviewer. The first objective of the study is to examine the main hydrometeorological processes of the 12-13 September 2019 widespread flash flooding. As consequence of the sustained rains, most of the examined catchments featured a dampened and delayed hydrological response: Until runoff thresholds were exceeded, infiltration-excess runoff generation did not start. As cumulative precipitation arises as the dominant flash flood-triggering mechanism, the study proposes a second objective: to explore simple scaling theory between flood magnitude and total rainfall amount from previous research. Admittedly, the reviewer is right when he/she stands that the final objective of using simple scaling theory between rainfall amount and flood magnitude should be to assess hydrological risk in arid and semi-arid basins. However, this work is just a first step in this research line. First, a power-law relationship between total rainfall amount and flood magnitude is analytically derived from previous research. Next, its potentialities are outlined through numerical experimentation. Future research should account for observed series of rainfall and discharge to empirically derive the envelope curve of a given catchment as basis for flood risk management. This goal is out of the present scope of the study and it is left for future research. In this sense, the second objective has been better contextualized in the revised manuscript (lines 18-22, 105-110, 530-532 in red).

**Specific comments:**

**L 375 Has the precipitation-discharge relationship been applied to observed data of the event?**

Figure 10 shows the observed specific peak discharges versus radar-derived total precipitation amounts (black rhombus) and how it compares with the quantitative precipitation forecasts.

**L 415 How can the power law relationship be used to perform analysis for different return periods? Maybe using synthetic rainfall coming from intensity duration frequency curves? Is the precipitation return period equal to peak discharge return period in a highly non linear response basin like the semiarid karst ones presented in the paper?**

Ideally, the envelope curves must be built from the available observed rainfall amounts and associated discharges in a catchment. These rainfall amounts would comprise different return periods if observed series are long enough. The corresponding discharges would be derived from the empirical envelope curves. Synthetic rainfall coming from intensity-duration-frequency curves could be used once the envelope curve has been built to derive the associated discharges.

No, there is not a direct relationship between the precipitation and discharge return periods in semi-arid and arid basins with karstic substrates as hydrological response is also highly dependent on initial soil moisture conditions. As stated in the conclusions, future research must use observations to establish the empirical envelope curves accounting for properties in rainfall (i.e., intensity and type) and initial soil moisture conditions (i.e., seasonality) for a given basin. All these questions still remain open at this research step.

**Technical corrections**

**L 102 "areat"**

Done

**REVIEWER 2**

**The manuscript "Hydrometeorological analysis of the 12 and 13 September 2019 widespread flash flooding in eastern Spain" by Arnaul Amengual presents an analysis of precipitation and flood response in several catchments impacted by a long-lasting heavy precipitation episode in September 2019, and in addition explores a simple scaling theory between flood magnitude and total rainfall amount. The manuscript reads well, but its structure is rather unusual. The author, after the introduction and case study section divides the manuscript into three sections (precipitation analysis, analysis of flood response and scale dependency between flood magnitude and rainfall amount), in which of each the methodology is provided and results presented. I would suggest to re-organize the paper in a more traditional structure, such that all the methodology is presented as first, followed by a section devoted to results presentation and discussion and a conclusive section in the end. Moreover, results should be inserted in a broader framework, highlighting differences/similarities with studies focusing on similar topics, which is rather missing in the current version of the manuscript.**

The author would like to thank the highly constructive comments and suggestions made by the reviewer. His/her concerns and suggestions have contributed to improve the structure and contents of the revised manuscript. As consequence, the structure of the manuscript has been restructured. The revised version follows a more classical standard. New section 3 is divided in 3 subsections presenting the methods used: 3.1) the spatial moments; 3.2) the hydrological model and calibration and; 3.3) the analytical framework of the scale dependency. Results and discussion are presented in the new section 4, following the same order than the methods. The revised manuscript finishes with the conclusive section 5. In addition, the revised version of the manuscript has been better contextualized in the framework of previous research in flash flooding over the Western Mediterranean, highlighting differences/similarities with previous studies focusing on similar topics. The new statements linked to this issue are highlighted in red throughout the revised manuscript.

**Please find more detailed comment below.**

**L57: There is plenty of studies about the use of the Generalized Extreme Value distribution, maybe the author could cite here some of the most relevant. The work by Metzger et al. (2020) was conducted with a specific focus on arid areas and concludes that the GEV and the partial duration series approach through the Generalized Pareto distribution are not optimal in the analyzed area. In the discussion, instead, they suggest to investigate the use of an emerging approach, i.e., the Metastatistical Extreme Value distribution in its full or simplified form (Marani and Ignaccolo (2015); Zorzetto et al. (2016); Marra et al. (2019); Miniussi et al. (2020) and references therein), as it leverages the use of the information contained by the whole distribution of streamflow values (in contrast to the largest ones only), is flexible in the choice of the underlying "ordinary distribution", thus allowing for a more accurate description of the tail of the distribution (e.g., Mushtaq et al., 2021) and was proved outperforming traditional methods especially in the case of short observational records, which are of interest for the present study.**

The author agrees with the reviewer concern. References about additional studies employing the Generalized Extreme Value distribution have been added in the revised version of the manuscript (lines 60-6a in red). Effectively, the Metastatistical Extreme Value (MEV) approach and its simplified form have arisen as suitable tools for dealing with the problematics of catchments featuring limited number of floods per year or limited data series, somewhat hampering the optimal application of the Generalized Extreme Value or Generalized Pareto distributions. A new paragraph has been included in the revised version of the manuscript to better contextualize the current status of flood frequency analysis in light of the reviewer's comments (lines 69-80 in red).

**L102-103: it would be helpful to see a map displaying for example the precipitation and temperature normals, so that it is immediately clear how their spatial pattern is.**

The author does not have available the necessary data to create these maps, although he agrees that they would be helpful. An intermediate compromise has been adopted by including a reference to a technical document by the Spanish Agency of Meteorology (AEMET) that presents this and other relevant climatological information of the study region (Chazarra-Bernabé et al. 2018)

**L133-138: this paragraph needs clarification: 1) some details about the "dynamical fitting" (which is not called like this in the Cole and Moore paper) would allow a deeper understanding on the procedure employed; 2) how are the 227 independent daily pluviometers selected? Are they a completely independent dataset or are they sub-sampled from the one including 369 gauges?**

The dynamic gauge-adjustments of radar estimates presented in Cole and Moore (2008) has been further clarified in the revised version of the manuscript (lines 151-154). Essentially, the implemented approach uses gridded multi-quadric surface fitting that vary in time and space. That is, the gauge-adjustment factors are calculated at the available radar temporal resolution.

The 227 independent daily pluviometers belong to the daily precipitation network of the Spanish Agency of Meteorology (AEMET). These stations perform daily measurements and constitute a different network from that of the automatic network of the same state body. The latter measures rainfall with a temporal resolution of 10 minutes, therefore, being completely independent. A sentence clarifying this issue has been added (line 155)

**L179 (and L433): how is the p-value of the OLS model computed?**

The estimates computed by the OLS regression are the mean of Gaussian random variables. Then, t-statistics are computed by using these estimates and their standard errors. Next, the p-value is obtained as the probability of achieving a t value as large as or larger than the observed t value is the null hypothesis was true. That is, it is computed the upper tail probability of achieving the t values that are obtained from a t distribution with degrees of freedom equal to the residual degrees of freedom of the model. In this case, the p-value represents the probability of achieving a t value greater than the absolute values of the observed ts.

**L343: what does "LNP" stand for? And why do you choose to evaluate the model performance by NSE and LNP? I suggest to add some comments on what they differ in, or the advantages/limitations of each one, so that their choice is supported.**

$L_{NP}$ is a skill measure of how well a model reproduces the observed system behaviour. It linearly combines various types of goodness-of-fit indices: the NSE efficiency coefficient and the error of peak time and peak discharge. Roux et al. (2011) called as the "$L_{NP}$" criterion, without any special meaning of the acronyms used. It was specifically designed to aid in warning decisions in emphasizing peak flow characteristics. That is, the $L_{NP}$ score attempts to conciliate real time flood forecasting requirements with a better understanding of the physical phenomena involved in flood event generation. The author agrees with the reviewer that in the context of the present study to add a second skill score is a bit redundant.

Therefore, the performance of the hydrologic simulations is evaluated by means of the Nash–Sutcliffe efficiency criterion and the relative errors in peak discharge and total direct runoff volume (lines 422-425). Table 3 has been modified accordingly

**Figures**

**Figure 9: what do the dashed lines represent?**

The horizontal dashed lines in Figure 9 have no special meaning. They have been introduced just for clarifying purposes when highlighting the different hydrological responses in terms of lag time. The vertical dash line in Figure 9(e) indicates whether rainfall distribution was located near the basin outlet or the headwaters.

**Typos/corrections**

Done